# Troponin elevation pattern and subsequent cardiac and non-cardiac outcomes: Implementing the Fourth Universal Definition of Myocardial Infarction and high-sensitivity troponin at a population level

Anthony (Ming-yu) Chuang[1,2]*, Mau T. Nguyen[1,2], Ehsan Khan[1,2], Dylan Jones[1,2], Matthew Horsfall[1], Sam Lehman[1,2], Nathaniel R. Smilowitz[3], Kristina Lambrakis[1,2], Martin Than[4], Julian Vaile[1,2], Ajay Sinhal[1,2], John K. French[5,6], Derek P. Chew[1,2]

1 College of Medicine and Public Health, Flinders University of South Australia, Adelaide, Australia, 2 Department of Cardiovascular Medicine, Southern Adelaide Local Health Network, Adelaide, Australia, 3 Division of Cardiology, Department of Medicine, New York University School of Medicine, New York City, New York, United States of America, 4 Department of Emergency Medicine, Christchurch Hospital, Christchurch, New Zealand, 5 South Western Sydney Clinical School, University of New South Wales, Sydney, Australia, 6 Western Sydney University, Sydney, Australia

* anthonychuang4@gmail.com

## Abstract

### Background

The Fourth Universal Definition of Myocardial Infarction (MI) differentiates MI from myocardial injury. We characterised the temporal course of cardiac and non-cardiac outcomes associated with MI, acute and chronic myocardial injury.

### Methods

We included all patients presenting to public emergency departments in South Australia between June 2011–Sept 2019. Episodes of care (EOCs) were classified into 5 groups based on high-sensitivity troponin-T (hs-cTnT) and diagnostic codes: 1) Acute MI [rise/fall in hs-cTnT and primary diagnosis of acute coronary syndrome], 2) Acute myocardial injury with coronary artery disease (CAD) [rise/fall in hs-cTnT and diagnosis of CAD], 3) Acute myocardial injury without CAD [rise/fall in hs-cTnT without diagnosis of CAD], 4) Chronic myocardial injury [elevated hs-cTnT without rise/fall], and 5) No myocardial injury. Multivariable flexible parametric models were used to characterize the temporal hazard of death, MI, heart failure (HF), and ventricular arrhythmia.

### Results

372,310 EOCs (218,878 individuals) were included: acute MI (19,052 [5.12%]), acute myocardial injury with CAD (6,928 [1.86%]), acute myocardial injury without CAD (32,231 [8.66%]), chronic myocardial injury (55,056 [14.79%]), and no myocardial injury (259,043 [69.58%]). We observed an early hazard of MI and HF after acute MI and acute myocardial

**Data Availability Statement:** The Human Research Ethics Committee of the South Australian Department of Health and Wellbeing provided approval to access all datasets to the authors (both identifiable and unidentifiable) and waived the requirement for individual participant informed consent. All those who access this data for this project have been required to enter into strict, legally-binding confidentiality agreements (this deed exists between the local Minister of Health and the Confidant Representative, which in this instance is Derek Chew, the senior author).The Ethics Committee stipulated that the datasets used by this study cannot be shared publicly as the data contain potentially identifying or sensitive patient information as per Australian National Statement guidelines (Section 2.3.6). As such, it is not appropriate from an ethical and legal perspective that this data is shared beyond those listed on the confidentiality deed for this dataset. Data could potentially be made available to individuals upon request, pending relevant agreements are put in place and approvals are granted. The following people can be contacted for data requests: Ming-yu Chuang (corresponding author, email provided with manuscript), Matthew Horsfall (clinical data manager, Health System Research, South Australian Health and Medical Research Institute, PO BOX 11060, Adelaide, South Australia, Australia, 5001. matthew.horsfall@health.sa.gov.au), and Marleesa Ly (project data manager, Health System Research, South Australian Health and Medical Research Institute, PO BOX 11060, Adelaide, South Australia, Australia, 5001. marleesa.ly@sahmri.com).

**Funding:** Dr Anthony (Ming-yu) Chuang is supported by the Royal Australasian College of Physicians' Fellows Research Entry grant. The funders had no role in study design, data collection and analysis, decision to publish, or preparation of the manuscript.

**Competing interests:** The authors have declared that no competing interests exist.

**Abbreviations:** ACS, Acute coronary syndrome; HR, Hazard ratio; MI, Myocardial infarction; CI, Confidence interval; Hs-cTnT, High-sensitivity cardiac troponin-T; EOC, Episodes of care; CAD, Coronary artery disease; eGFR, Estimated glomerular filtration rate; HF, Heart failure; CKD, Chronic kidney disease; NOFF, Neck of femur fracture; ED, Emergency department.

injury with CAD. In contrast, subsequent MI risk was lower and more constant in patients with acute injury without CAD or chronic injury. All patterns of myocardial injury were associated with significantly higher risk of all-cause mortality and ventricular arrhythmia.

## Conclusions

Different patterns of myocardial injury were associated with divergent profiles of subsequent cardiac and non-cardiac risk. The therapeutic approach and modifiability of such excess risks require further research.

## Introduction

Cardiac troponin is a highly sensitive biomarker for myocardial injury that plays an essential role in the diagnosis and risk-stratification of acute myocardial infarction (MI) [1, 2]. The improved analytical sensitivity of the new high-sensitivity cardiac troponin (hs-cTn) assays facilitates early diagnosis of MI. However, these assays come with new challenges including increased identification of troponin elevations above the conventional reference threshold (>99th percentile upper reference limit) in patients without objective evidence of myocardial ischemia (e.g. on echocardiography or ECG) [3–5].

The Fourth Universal Definition of MI is the first guideline to formally define this syndrome as myocardial injury and distinguish it from MI. It outlines three main patterns of troponin elevation—acute MI, acute myocardial injury and chronic myocardial injury [2]. Acute MI is defined as myocardial injury with clinical evidence of myocardial ischaemia and can be subdivided into five types: type 1 (atherosclerotic plaque rupture), type 2 (supply-demand mismatch), type 3 (cardiac death prior to availability of troponin results), type 4 (percutaneous coronary intervention related), and type 5 (cardiac surgery related) [2]. In contrast, myocardial injury is defined as an elevated troponin without evidence of myocardial ischaemia, and is subdivided into acute and chronic injury depending on the presence or absence of an observed rise and/or fall in troponin levels, respectively [2].

A number of studies have indicated that myocardial injury is now the most common cause of troponin elevation [4, 6, 7] and confers a poor prognosis independent of the underlying mechanism of its elevation [6, 8–11]. There is also evolving evidence to suggest that each of these different patterns of troponin elevation has distinct clinical consequences [12, 13]. For example, patients with acute myocardial injury and type 2 MI appear to have worse short-term [14] and long-term [12, 13] mortality compared with patients with type 1 MI [7, 13, 15–18]. However, the temporal association with subsequent cardiovascular complications, such as MI and heart failure (HF), are uncertain. Another critical difference between the diagnoses of myocardial injury and MI is the disparity in evidence to inform clinical management. While there is rich evidence to guide the management of type 1 MI, there is little evidence to guide the management of patients with type 2 MI and myocardial injury [19–21]. This is especially pertinent as type 1 MI accounts for a relatively small proportion of all detectable troponin elevations and myocardial injury is increasingly observed in clinical practice [4, 6, 10].

The short- and long-term consequences of different classifications and patterns of troponin elevation may provide crucial insights into the design of future clinical trials to test interventions to treat myocardial injury without MI. The aim of the present study was to characterise the temporal hazard of cardiac and non-cardiac events associated with the different classifications and patterns of troponin elevation using population-level data from a large health system in Australia.

## Methods

### Study population and diagnostic classifications

We identified consecutive patients presenting to public hospital emergency departments (EDs) in South Australia between July 2011 to September 2019 who had at least one high-sensitivity cardiac troponin-T (hs-cTnT) measured during their ED stay. In July 2011, a 5th generation hs-cTnT assay was implemented across all public hospitals in the state by a single pathology service, with the same assay implemented at all facilities. Troponin results were linked to hospital records and International Classification of Diseases primary and secondary diagnostic codes, version 10 Australian Modified (ICD-10 AM). Each encounter was considered as a new episode of care (EOC). Each EOC for a given patient was linked longitudinally allowing representation to hospital to serve both as an outcome for the prior EOC as well as a new EOC. Transfers between hospitals were considered as part of the same EOC. All diagnostic codes for patients transferred between hospitals were evaluated to ensure all suitable cases were identified. Episodes of care were excluded from the analyses if troponin testing was not performed or if there was only a single borderline elevated hs-cTnT (29-52ng/L) since a troponin pattern could not be determined from these episodes of care based on published studies [22–25]. Patients admitted through ED for elective coronary artery bypass surgery were also excluded due the expected differences in subsequent prognosis. Each EOC was followed up for a minimum of 12 months and was censored at the time of last known follow-up. The decision for all clinical management was made at the treating physician's discretion, independent of this study.

Trained independent coding professionals, applying standardized audited protocols, used medical record documentation, imaging and pathology data to classify primary and secondary diagnoses for each clinical presentation. Within the current coding conventions, the diagnoses listed as the "primary diagnosis" were deemed to be the main reason for which the patient presented for clinical attention. "Secondary diagnoses" represent those conditions recognized to impact the complexity of subsequent clinical care. Where more than one cardiac diagnostic code was present, the primary diagnostic code was used. Significant past medical conditions were determined by examining hospitalization records from the preceding 10 years. Deaths and their cause were identified through hospital records and the State Death Registry.

The study population was identified based on the above inclusion and exclusion criteria and was not anonymized during the data linkage process, although the final dataset was fully anonymized. As part of the de-identification process, a unique identifier was created and assigned to each patient by the data manager to allow for identification of reattendances. The unique identifier obviates the risk of re-identification and was only accessible by the data manager. After a unique identifier was assigned to each participant, the dataset was then fully de-identified for analysis purposes. The need for patient consent was waived by the local ethics committee as there were robust procedures to ensure sufficient protection of patient data as per Australian National Statement guidelines (Section 2.3.6), the study involved negligible risk to the study participants and there is adequate plan to ensure the ongoing confidentiality of the data. The Human Research Ethics Committee of the South Australian Department of Health and Wellbeing provided approval to access all datasets described above and this study complies with the Declaration of Helsinki (HREC/19/SAH/36).

### Biomarker measurements

The indication and timing for hs-cTnT testing was clinically determined. All troponin samples were analysed using 5th generation hs-cTnT assay (Roche Diagnostics: lower limit of quantification: 5ng/L; 99th percentile upper reference limit in a normal population: male = ≥22ng/L,

female = $\geq$14ng/L; lowest concentration with a coefficient of variation <10%: 4.49 ng/L). During the study period, an elevated troponin was only reported to the treating clinician at a hs-cTnT concentration $\geq$29ng/L due to previous and ongoing studies [26, 27]. A clinically significant rise and/or fall was determined as a relative change of $\geq$20% or a gradient of $\geq$3ng/L/hr from initial to any subsequent measurement within 24 hours as per expert consensus and existing literature [2, 28].

## Group definitions

We classified each EOC into five groups based on their troponin pattern and primary/secondary discharge diagnoses to maximally align with the Fourth Universal Definition of MI. Episodes of care rather than individual patient-level analysis was chosen as pattern characterisation of hs-cTnT allows for dynamic risk assessment which is expected to vary in an individual over time. Additional sensitivity analysis was also performed excluding patients with very high frequency of non-cardiovascular hospital presentations ($\geq$4 ED presentations per year unrelated to the outcomes of interest).

**1) Acute MI.** Defined as an EOC with at least one elevated hs-cTnT measurement above 99th centile (male: $\geq$22ng/L, female: $\geq$14ng/L) with a qualifying rise and/or fall (relative change of $\geq$20% or gradient of $\geq$3ng/L/hr within the first 24 hours of presentation) and a primary diagnostic code of coronary artery disease (CAD) (ICD-10-AM codes I20-I25). In addition, EOC were also classified as acute MI if the following criteria were fulfilled: a) patients who died within 12 hours of presentation with at least one elevated hs-cTnT and had a primary diagnosis of MI (ICD-10-AM of I21), b) patients who had a single hs-cTnT $\geq$52ng/L and with a primary diagnosis of MI (ICD-10-AM of I21) if only one hs-cTnT measurement was performed, and c) patients who had a hs-cTnT >250ng/L without an observed rise and/or fall with a primary diagnosis of MI (ICD-10-AM of I21). The hs-cTnT cut-off value of $\geq$ 52ng/L was chosen based on published studies [22–25].

**2) Acute myocardial injury with recognized CAD.** Defined as an EOC with at least one elevated hs-cTnT measurement with a qualifying rise and/or fall and a secondary diagnostic code associated with CAD (ICD-10-AM codes I20-I25), but where the primary diagnostic code was not due to an acute coronary syndrome. This classification implied an underlying supply-demand ischaemia mechanism as an approximation of Type 2 MI but does not always exclude a diagnosis of acute type 1 MI.

**3) Acute myocardial injury without recognized CAD.** Defined as an EOC with at least one elevated hs-cTnT measurement with a qualifying rise and/or fall without any diagnostic codes associated with CAD or previously known CAD diagnosis.

**4) Chronic myocardial injury.** Defined as an EOC with at least one elevated hs-cTnT measurement without a qualifying rise and/or fall.

**5) No myocardial injury.** Defined as an EOC with no hs-cTnT above 99th centile.

To mitigate against misclassification bias associated with using administrative data, 6,362 EOC were adjudicated by two independent clinicians and the classification criteria were optimised using the adjudicated diagnoses. The final optimised classification criteria had an accuracy of 85.35% (S2 Table). The majority of the misclassification was due to borderline hs-cTnT results, borderline hs-cTnT changes (e.g. female patient with hs-cTnT level of 13.9 ng/L, relative hs-cTnT change of 19%), and the presence of uncoded CAD.

## Outcomes of interest

The primary outcomes assessed were times to all-cause mortality, new/recurrent acute MI and associated complications (I21-I23), HF admissions (I42, I43, I255, I50, J81) and the composite

outcome of ventricular arrhythmias and cardiac arrest (I46, I47.0, I47.2, I47.9). Acute MI was chosen as it infers an underlying coronary atherosclerosis as a substrate of subsequent risk, which may be modified by coronary-specific investigations and therapies (e.g. HMG-CoA reductase inhibitors, antiplatelet agents and anatomical assessment +/- coronary revascularisation). HF admission and ventricular arrhythmia/cardiac arrest were chosen as they may represent medium to long-term consequences of myocardial damage. Time to admission for pneumonia and neck of femur fracture (NOFF) were chosen as secondary outcomes of interest as these non-cardiac clinical events may provide insights into systemic burden of co-morbidity and frailty, respectively.

## Statistical analysis

Continuous variables were tested for normal distribution and were reported either as means with standard deviation or as medians with 25th and 75th percentiles. Categorical variables were reported as frequencies and proportions. Baseline characteristics were compared using Pearson's chi-square test for categorical variables and analysis of variance or Kruskal-Wallis test for continuous variables where appropriate.

To estimate the excess hazard associated with different types of MI and myocardial injury, multivariable flexible parametric models with time-varying covariates and restricted cubic splines (varying spline knots) with utilized and EOC with no myocardial injury as the comparator [29, 30]. The selection of the number of internal spline knots in the Royston and Parmar (RP) model was guided by optimizing the Akaike information criterion. The proportional hazards scale was used in the RP model to facilitate comparison of the hazard ratios (HRs) observed. Estimates are reported as HRs with 95% confidence intervals (95% CI). Factors considered as potential confounders were age in years, sex, lowest in-hospital estimated glomerular filtration rate (eGFR), maximal in-hospital hs-cTnT, clinical comorbidities such as diabetes mellitus, chronic obstructive pulmonary disease, dementia, peripheral artery disease, and previous stroke. A sensitivity analysis was performed excluding patients with very high frequency of non-cardiovascular hospital presentations ($\geq$4 ED presentations per year unrelated to the outcomes of interest).

In a secondary analysis, the consequence of repeated injury associated with each type of myocardial injury was assessed by including a cumulative count variable of the different myocardial injury types in Cox regression models (e.g. 1, 2, or $\geq$ 3 prior presentations with chronic myocardial injury). The model used for this analysis corrected for previous MI or myocardial injury types, as a patient may experience more than one type of myocardial injury during the follow-up period (e.g. 2 prior presentation of chronic myocardial injury and 1 prior presentation of acute myocardial injury), and other clinical covariates included in the flexible parametric models. The measured outcomes from this analysis were the HRs of subsequent acute MI and HF admissions. For this analysis, acute MI and acute myocardial injury with recognized CAD were merged and treated as one group as they both have CAD as a substrate of subsequent ischaemic risk. All reported p-values were 2-sided, and statistical significance was set at p<0.05. All analyses were performed using STATA 16.1 (College Station TX, USA).

## Results

### Patient characteristics

Cohort selection is outlined in **Fig 1**. Between June 2011 and September 2019, 372,310 EOC (218,878 individual patients) out of a total of 1,595,725 EOC (246,381 individual patients) met the inclusion criteria and were included in the analysis (**Fig 1**). Episodes of care were classified into five groups based on the a priori defined criteria: 1) Acute MI (n = 19,052, 5.12%), 2) Acute myocardial injury with recognized CAD (n = 6,928, 1.86%), 3) Acute myocardial injury

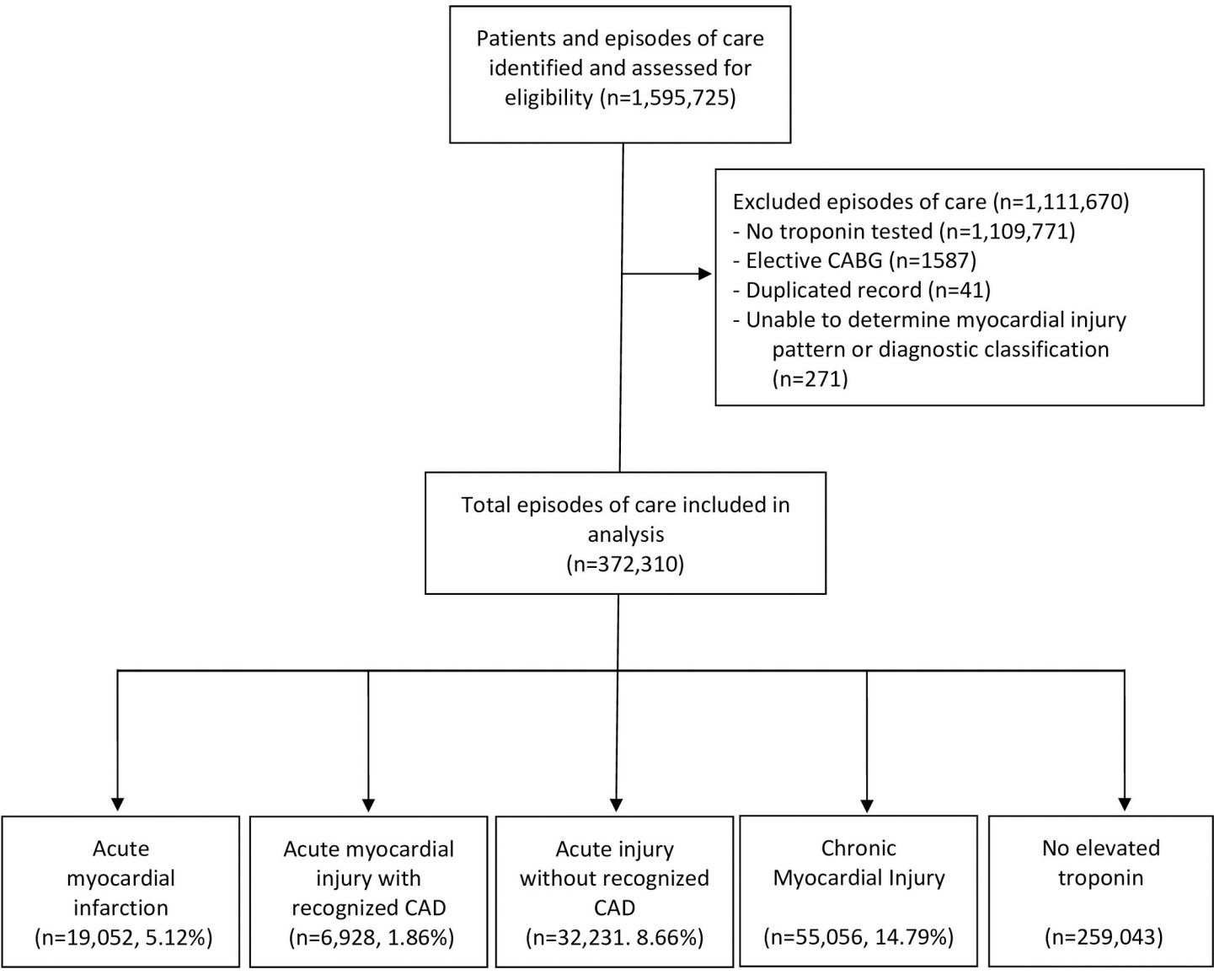

**Fig 1. Study flow diagram.** CAD = Coronary artery disease. CABG = coronary artery bypass graft.

without recognized CAD (n = 32,231, 8.66%), 4) Chronic myocardial injury (n = 55,056, 14.79%), and 5) No myocardial injury (n = 259,043, 69.58%). The clinical characteristics of patients in each group are presented in **Table 1**. Overall, acute MI accounted for 16.8% of all EOCs with hs-cTnT elevation and this group had the highest level of peak hs-cTnT. Patients who had chronic myocardial injury were older, had lowest peak hs-cTnT levels, and greater burden of co-morbidities. Compared with patients who had acute myocardial injury with CAD, patients who had acute myocardial injury without recognized CAD were of comparable age but had lower peak hs-cTnT levels.

## Short- and long-term outcomes

The observed incidence of all-cause mortality, subsequent MI, HF admission, and ventricular arrhythmias or cardiac arrest at 1-month, and 1-year are presented in **Table 2**. The unadjusted

**Table 1. Patient characteristics.**

| | Acute myocardial infarction (n = 19,052) | Acute myocardial injury with recognized CAD[a] (n = 6,928) | Acute myocardial injury without recognized CAD (n = 32,231) | Chronic myocardial injury (n = 55,056) | No myocardial injury (n = 259,043) | P-value |
|---|---|---|---|---|---|---|
| Age (years, median, i.q.r[b]) | 70 (58–81) | 78 (68–86) | 77 (66–90) | 81 (72–87) | 56 (43–69) | <0.001 |
| Female (n, %) | 6,231 (32.71) | 2,853 (41.18) | 15,867 (49.23) | 24,811 (45.07) | 130,428 (50.36) | <0.001 |
| 1-year all-cause mortality (n, %) | 533 (2.80) | 325 (4.69) | 945 (2.93) | 772 (1.40) | 481 (0.19) | <0.001 |
| 1-year myocardial infarction (n, %) | 244 (1.28) | 28 (0.4) | 87 (0.27) | 151 (0.27) | 127 (0.05) | <0.001 |
| Maximum 24-hour hs-cTnT[c] (ng/L, median, i.q.r) | 338 (100–1159) | 212 (70–524) | 50 (28–114) | 39 (25–63) | 6 (3–9) | <0.001 |
| eGFR[d] (mLs/min/1.73m$^2$, median, i.q.r) | 71 (52–88) | 53 (34–73) | 61 (41–82) | 54 (36–74) | 90 (76–106) | <0.001 |
| Known diabetes mellitus (n, %) | 4,524 (23.75) | 2,415 (34.86) | 8,909 (27.64) | 19,492 (35.40) | 30,130 (11.63) | <0.001 |
| Known COPD[e] (n, %) | 1,497 (7.86) | 1,255 (18.11) | 5,456 (16.93) | 10,805 (19.63) | 13,852 (5.35) | <0.001 |
| Known PAD[f] (n, %) | 1,371 (7.20) | 836 (12.07) | 2,794 (8.67) | 6,289 (11.42) | 6,089 (2.35) | <0.001 |
| Prior CVA[g] (n, %) | 850 (4.46) | 543 (7.84) | 2,232 (6.93) | 4,687 (8.51) | 6,913 (2.67) | <0.001 |
| Known dementia (n, %) | 455 (2.39) | 342 (4.94) | 1,587 (4.92) | 3,488 (6.34) | 2,552 (0.99) | <0.001 |
| Known prior heart failure | 2278 (11.96) | 1999 (28.85) | 7156 (22.20) | 16622 (30.19) | 12234 (4.72) | <0.001 |
| Known prior ventricular arrhythmia | 317 (1.66) | 288 (4.16) | 830 (2.56) | 1808 (3.29) | 3344 (1.29) | |
| Coronary angiography during index presentation | 12469 (64.45) | 1787 (25.79) | 1295 (4.02) | 2467 (4.48) | 5696 (2.20) | |

[a]CAD = coronary artery disease

[b]i.q.r. = interquartile range'

[c]hs-cTnT = high-sensitivity troponin-T

[d]eGFR = estimated glomerular filtration rate

[e]COPD = chronic obstruction pulmonary disease

[f]PAD = peripheral artery disease

[g]CVA = cerebral vascular disease.

1-year mortality following EOCs with acute myocardial injury with CAD was higher than those with acute MI (acute MI: 533/19052 [2.80%], acute injury with CAD: 325/6928 [4.69%], acute injury without CAD: 945/32231 [2.93%]; overall p-value: <0.001). The risk for subsequent MI was highest following acute MI, although the risk following acute myocardial injury with recognized CAD was also high. The observed 1-year incidence of pneumonia and NOFF were similar across all groups (**S1 Table**). Using the flexible multivariable parametric models, the adjusted temporal HRs for all-cause mortality, subsequent MI, HF admission and ventricular arrhythmias or cardiac arrest for the four groups were modelled (**Fig 2, Table 3, S1 Fig**). Distinct patterns of risk were observed for each group. Among patients with acute MI, we observed an early hazard for recurrent MI that slowly declined over time. In contrast, patients with acute myocardial injury with recognized CAD had a more constantly elevated hazard of future MI whilst patients without recognized CAD (either acute or chronic injury pattern) had a lower risk of future MI (adjusted 1-year HR of MI in acute myocardial injury with CAD group: HR = 2.6, 95% confidence interval [95%CI] = 2.25–3.04; acute myocardial injury without CAD group: HR = 1.51, 95%CI = 1.33–1.72; chronic myocardial injury group: HR = 1.95,

**Table 2. Observed incidence of all-cause mortality, subsequent myocardial infarction, heart failure admission and ventricular arrhythmia or cardiac arrest at 1-month, and 1-year.**

| | All-cause mortality | Subsequent MI | HF admission | Ventricular arrhythmia/ cardiac arrest |
|---|---|---|---|---|
| **Acute myocardial infarction (n = 19,052)** | | | | |
| 1-month (n, %) | 178 (0.93) | 4 (0.02) | 1 (0.01) | 0 (0) |
| 1-year (n, %) | 533 (2.80) | 244 (1.28) | 103 (0.54) | 4 (0.02) |
| **Acute myocardial injury with recognized CAD (n = 6,928)** | | | | |
| 1-month (n, %) | 71 (1.02) | 0 (0) | 0 (0) | 0 (0) |
| 1-year (n, %) | 325 (4.69) | 28 (0.40) | 51 (0.74) | 5 (0.07) |
| **Acute myocardial injury without recognized CAD (n = 32,231)** | | | | |
| 1-month (n, %) | 262 (0.81) | 10 (0.03) | 4 (0.01) | 0 (0) |
| 1-year (n, %) | 945 (2.93) | 87 (0.27) | 163 (0.51) | 11 (0.03) |
| **Chronic myocardial injury (n = 55,056)** | | | | |
| 1-month (n, %) | 99 (0.18) | 6 (0.01) | 10 (0.02) | 1 (<0.01) |
| 1-year (n, %) | 772 (1.40) | 151 (0.27) | 419 (0.76) | 18 (0.03) |
| **No myocardial injury (n = 259,043)** | | | | |
| 1-month (n, %) | 137 (0.05) | 8 (<0.01) | 8 (<0.01) | 1 (<0.01) |
| 1-year (n, %) | 481 (0.19) | 127 (0.05) | 172 (0.07) | 32 (0.01) |

CAD = coronary artery disease, MI = myocardial infarction, HF = heart failure.

95% = 1.79–2.13). We also observed that all patterns of myocardial injury were associated with an early risk of all-cause mortality with acute injury (with or without CAD) at the greatest risks. Both acute MI and acute myocardial injury with recognised CAD were associated with an early hazard of HF, whereas the other two groups had a more constantly elevated hazard of subsequent HF. All groups were associated with excess risk of ventricular arrhythmias or cardiac arrest, with acute myocardial injury with recognized CAD at the highest risk (**Fig 2**, **S1 Fig**). Excluding multiple repeat non-cardiovascular presentations (≥4 ED presentations per year unrelated to the outcomes of interest) did not significant change the overall pattern of results (**S3 Table**).

## Effect of recurrent injury types

To assess the consequences of repeated myocardial injury or infarction, a 'cumulative injury count' variable was included in Cox regression models with the type of injury subdivided into three groups: 1) acute myocardial injury with CAD (pooled acute MI and acute myocardial injury with recognized CAD), 2) acute myocardial injury without recognized CAD and 3) chronic myocardial injury (**Fig 3**, **S4 Table**).

Based on the flexible multivariable model, a strong dose-response relationship was observed between the number of episodes of acute myocardial injury with recognized CAD and subsequent MI risk. In contrast, the association between cumulative episodes of acute myocardial injury without recognized CAD or chronic myocardial injury and the subsequent risk of MI were not as pronounced (**Fig 3A**, **S4 Table**). All three groups were associated with an incremental increase in risk of subsequent HF admission (**Fig 3B**, **S4 Table**).

## Discussion

The Fourth Universal Definition of MI attempts to simplify the complexities in the interpretation and management of different patterns of cardiac troponin elevation by defining three distinct phenotypic patterns (acute MI, acute myocardial injury and chronic myocardial injury)

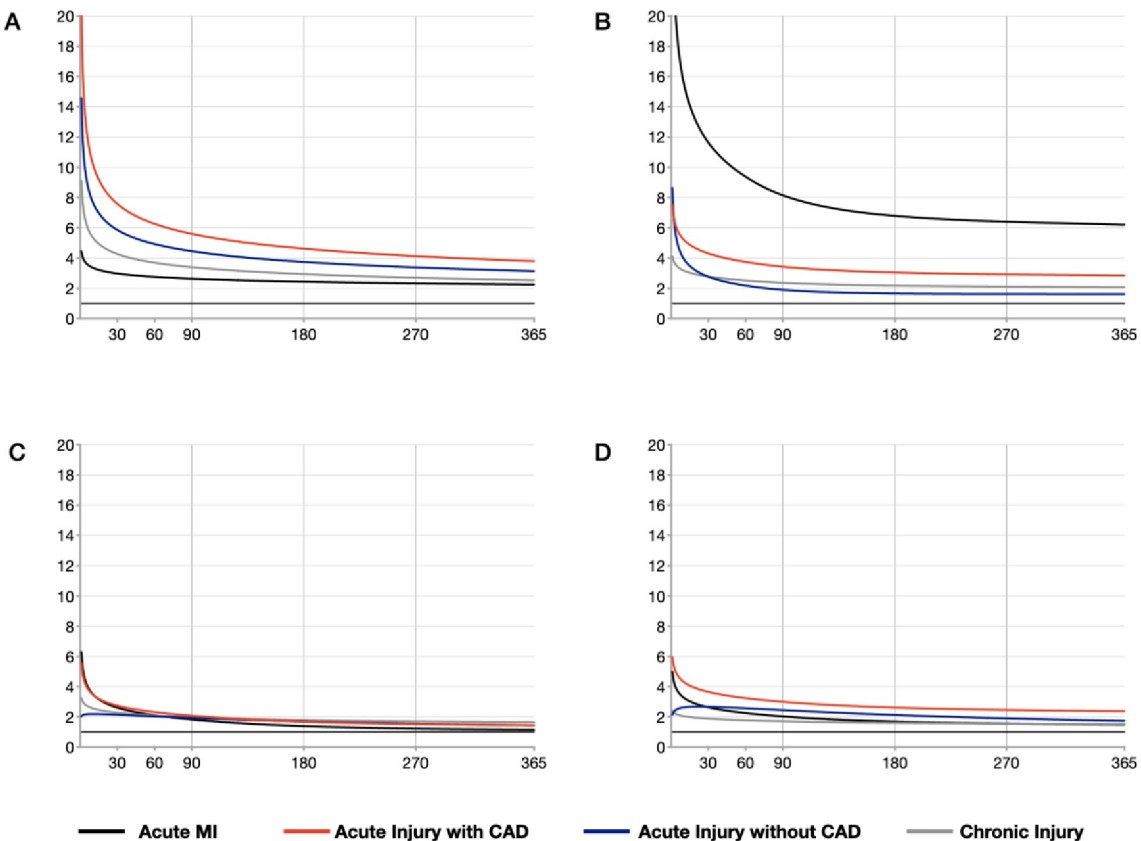

**Fig 2.** Estimated hazard ratio of A) all-cause mortality, B) subsequent myocardial infarction, C) subsequent heart failure admission and D) ventricular arrhythmia or cardiac arrest in patients with acute myocardial infarction, acute myocardial injury with recognized coronary artery disease, acute myocardial injury without recognized coronary artery disease and chronic myocardial injury, relative to patients with no myocardial injury. Graphs were adjusted for age in years, sex, lowest in-hospital estimated glomerular filtration rate, maximal in-hospital high-sensitivity troponin-T, clinical comorbidities including diabetes mellitus, chronic obstructive pulmonary disease, dementia, peripheral artery disease, and previous stroke. MI = myocardial infarction. CAD = coronary artery disease.

[2]. While there exists a robust evidence-base to guide management and risk stratification of acute MI, especially for type 1 acute MI (atherosclerotic plaque disruption), there is a lack of guiding evidence for prognosis-modifying therapies for patients with acute and chronic myocardial injury [20, 21]. In this population-level cohort study, we characterized the temporal risk associated with different troponin elevation patterns and subtypes and examined the clinical outcomes after repeated presentations of myocardial injury or infarction. The primary findings of this study were: 1) the divergent profiles of subsequent risk amongst the different phenotypes of myocardial injury, 2) additive risk for subsequent MI in repeat presentations of acute myocardial injury with recognized CAD, 3) the value of troponin as a cardiac-selective risk stratification tool in patients with high burden of competing non-cardiac comorbidities, and 4) the high proportion of non-MI hs-cTnT elevations. To our knowledge, this is the first health system-level study to evaluate the individual temporal clinical consequences associated with different phenotypes of myocardial injury using a contemporary high-sensitivity troponin assay. Our study should inform the design of future clinical trials of current and emerging therapies that may impact cardiac outcomes among these high-risk groups currently not served by an evidence base to guide therapy.

Our observation of distinct temporal profiles of subsequent clinical events may provide insights into potential underlying pathophysiology for each phenotype of troponin elevation

**Table 3. Adjusted hazard ratios of 30-day, 1-year and 3-year all-cause mortality, recurrent myocardial infarction, subsequent heart failure admission, and ventricular arrhythmias or cardiac arrest in the acute myocardial infarction, acute myocardial injury with recognized coronary artery disease, acute myocardial injury without recognized coronary artery disease and chronic myocardial injury groups based on flexible parametric models.**

| | All-cause mortality | Recurrent MI | HF admission | Ventricular arrhythmia/ cardiac arrest |
|---|---|---|---|---|
| **Acute myocardial infarction** | | | | |
| **30-day HR (95%CI)** | 2.98 (2.04–4.33) | 9.04 (7.96–10.28) | 2.61 (2.26–3.00) | 2.63 (2.03–3.42) |
| **1-year HR (95%CI)** | 2.25 (1.95–2.60) | 5.87 (5.38–6.41) | 1.15 (1.03–1.27) | 1.48 (1.21–1.82) |
| **3-year HR (95% CI)** | 1.87 (1.76–2.00) | 4.84 (34.3–5.42) | 0.90 (0.79–1.02) | 1.30 (1.09–1.53) |
| **Acute myocardial injury with recognized CAD** | | | | |
| **30-day HR (95%CI)** | 7.58 (5.19–11.07) | 3.34 (2.59–4.31) | 2.74 (2.30–3.27) | 3.65 (2.62–5.12) |
| **1-year HR (95%CI)** | 3.80 (3.28–4.40) | 2.61 (2.25–3.04) | 1.44 (1.30–1.60) | 2.37 (1.90–2.95) |
| **3-year HR (95%CI)** | 3.04 (2.75–3.36) | 2.00 (1.61–2.49) | 1.16 (1.02–1.33) | 1.93 (1.54–2.48) |
| **Acute myocardial injury without recognized CAD** | | | | |
| **30-day HR (95%CI)** | 5.86 (4.29–8.00) | 2.21 (1.80–22.71) | 2.14 (1.89–2.42) | 2.67 (2.08–3.42) |
| **1-year HR (95%CI)** | 3.14 (2.80–3.51) | 1.51 (1.33–1.72) | 1.44 (1.33–1.56) | 1.74 (1.45–2.09) |
| **3-year HR (95%CI)** | 2.29 (2.16–2.43) | 1.44 (1.24–1.67) | 1.20 (1.11–1.31) | 1.35 (1.12–1.68) |
| **Chronic myocardial injury** | | | | |
| **30-day HR (95%CI)** | 4.27 (3.32–5.49) | 2.36 (2.02–2.745) | 2.27 (2.05–2.52) | 1.88 (1.49–2.35) |
| **1-year HR (95%CI)** | 2.53 (2.30–2.79) | 1.95 (1.79–2.13) | 1.66 (1.57–1.75) | 1.52 (1.32–1.76) |
| **3-year HR (95%CI)** | 2.03 (1.92–2.15) | 1.70 (1.51–1.92) | 1.46 (1.37–1.56) | 1.45 (1.32–1.66) |

Flexible parametric model adjusted for age in years, sex, lowest in-hospital estimated glomerular filtration rate, maximal in-hospital high-sensitivity troponin-T, clinical comorbidities including diabetes mellitus, chronic obstructive pulmonary disease, dementia, peripheral artery disease, and previous stroke. CAD = coronary artery disease, HR = hazard ratio, 95%CI = 95% confidence interval, MI = myocardial infarction, HF = heart failure.

assessed (**Fig 2**, **S1 Fig**). The acute MI group showed an expected pattern consistent with acute plaque rupture, with an early elevated risk for mortality, MI, heart failure and ventricular arrhythmias, which diminishes over time. Therapies with the likely greatest risk-modifying potential for this group are therefore acute coronary-specific therapies in keeping with current guidelines (i.e. antiplatelet therapies, early non-invasive or invasive anatomical assessment +/- revascularisation, and HMG-CoA reductase inhibitors) [19–21]. In contrast to acute MI, the myocardial injury groups did not show the pattern of early elevated ischaemic risk. Instead, the elevated ischaemic risk was only seen in patients with documented CAD and was more constant over time (**Fig 2**, **S1 Fig**). Prior studies that have characterised outcomes following non-type 1 MI have been limited by relatively small sample sizes or have combined clinical endpoints such as recurrent MI and heart failure, thus making it difficult to propose putative mechanisms for poor long-term outcomes [7, 13, 16]. Consistent with these studies [7, 13, 16], we observed a higher risk of long-term all-cause mortality in myocardial injury than in MI. It remains unclear whether this higher mortality in myocardial injury is due to fundamental differences in the mechanism(s) of troponin elevation, the influence of comorbidities, or the effect of disease-specific treatment available. Our observation that patients with acute myocardial injury without recognized CAD had higher all-cause mortality despite having lower subsequent ischaemic risk suggests that these patients may not benefit from therapies targeted at acute plaque rupture and may in fact be harmed by them due to the concomitant bleeding risk. This may be especially relevant during the Coronavirus-19 pandemic where acute myocardial injury is common and bleeding risk is high [31–34]. Overall, we observed a lower 1-year mortality in the chronic myocardial injury group compared to other studies [7, 12, 16], although after adjustment, the mortality risk of this group was comparable to the acute MI group (adjusted 1-year hazard ratio of mortality for chronic injury vs. acute MI: 2.53 vs. 2.25).

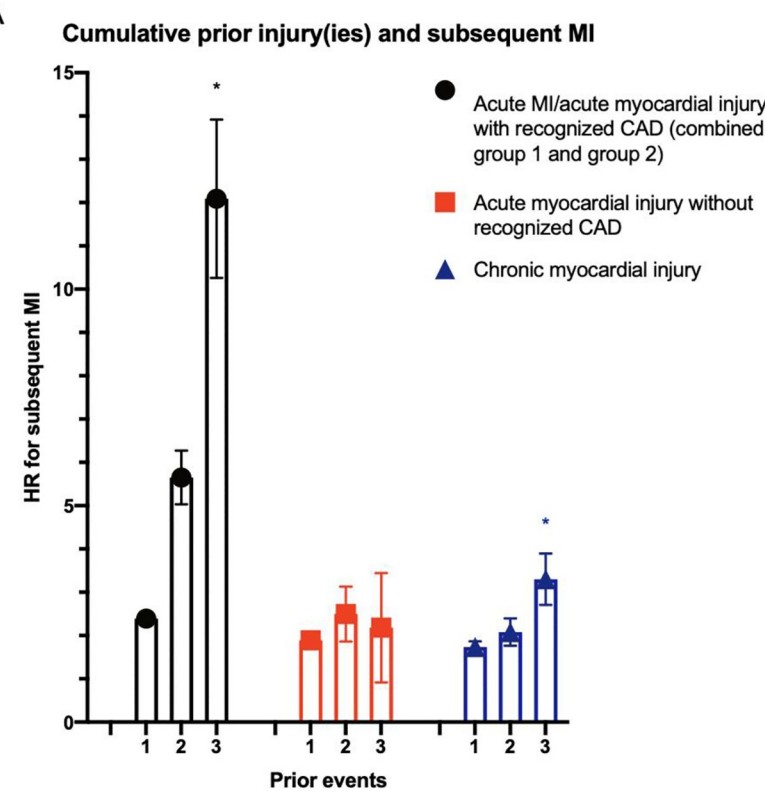

A

**Cumulative prior injury(ies) and subsequent MI**

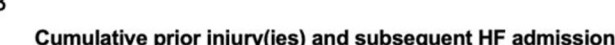

B

**Cumulative prior injury(ies) and subsequent HF admission**

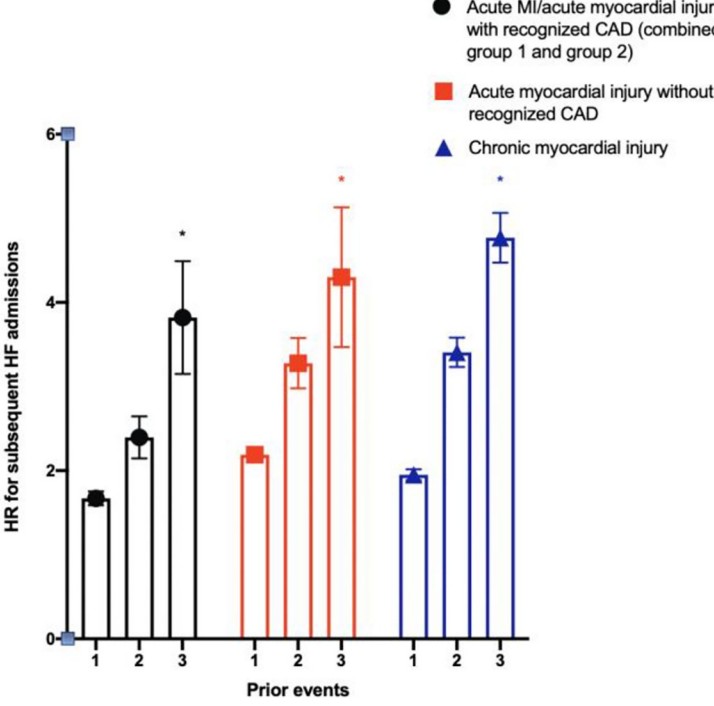

**Fig 3.** Estimated hazard ratio of A) subsequent myocardial infarction(s) and B) subsequent heart failure admission(s) in patients with 1, 2, and ≥3 recurrent episodes of acute myocardial injury with recognized coronary artery disease (combined group 1 and 2), acute myocardial injury without recognized coronary artery disease (group 3) and chronic myocardial injury (group 4). Graphs were adjusted for adjusted for age in years, sex, lowest in-hospital estimated glomerular filtration rate, maximal in-hospital high-sensitivity troponin-T, clinical comorbidities including diabetes mellitus, chronic obstructive pulmonary disease, dementia, peripheral artery disease, and previous stroke. model. CAD = coronary artery disease, HF = heart failure, MI = myocardial infarction. * denotes statistically significant trend with p-value <0.05.

The relatively low unadjusted mortality may be partially explained by the fact that our analysis focused on characterization of patients' risk "per encounter" and patients who were initially classified as chronic myocardial injury may subsequently experience an 'acute injurious' event that preceded mortality and therefore the mortality risk was attributed to acute injury. Lastly, we found no association between myocardial injury (both acute and chronic) and markers of frailty (subsequent NOFF) or susceptibility to illness (subsequent pneumonia) after correcting for potential confounders. Whilst previous studies have suggested that troponin elevation may be non-specific in patients with highly competing non-cardiac comorbidities [35–37], our findings suggest that troponin remains a useful tool for cardiac-selective risk stratification.

One of the key findings of this study was the observation of the large potentially modifiable risk for recurrent MI in patients with acute myocardial injury with recognized CAD, compared to the other two injury groups (acute myocardial injury without recognized CAD and chronic myocardial injury). In this study, we chose to additionally subdivide acute myocardial injury into patients with and without recognized CAD, under the hypothesis that these represent distinct mechanistic subgroups. In keeping with this hypothesis, we observed clear differences between these two subgroups especially in the risk of subsequent MI (**Fig 2**, **Table 3**, **S1 Fig**). Specifically, we observed a greater than 3-fold increase in risk of subsequent MI in patients with acute myocardial injury with recognized CAD that was not observed in those without recognized CAD. The difference between the early elevated risk in acute MI and the constantly elevated MI risk in acute myocardial injury likely represent a mechanistic difference in acute plaque rupture versus more stable coronary artery disease (**Fig 2**, **S1 Fig**). We also observed an incremental dose-response relationship between repeated acute myocardial injury with recognized CAD and subsequent ischaemic events (**Fig 3**). This dose-response relationship was not detected in patients with acute myocardial injury without recognized CAD or chronic myocardial injury, further supporting the concept of distinct underlying pathophysiologies between phenotypes. A previous study has also observed a difference between these subgroups, as CAD was shown to be an independent risk factor for major adverse cardiovascular events in patients with myocardial injury or type 2 MI [13]. Thus, our results would suggest the potential benefit of invasive or non-invasive anatomical assessment in patients with acute myocardial injury if coronary anatomy is unknown, followed by coronary-specific preventative therapies such as HMG-CoA reductase inhibitors to mitigate their risk for subsequent coronary events. This, of course, requires clinical judgement in the balancing of the risks associated with vascular access (if invasive) and contrast-induced kidney injury. Overall, whilst previous studies have demonstrated an association between troponin elevation and poor outcome [7, 13, 15–18], our findings suggest that each pattern of troponin elevation likely necessitate distinct risk-modifying strategies to mitigate these outcomes. These concepts of coronary investigation and directed therapies require further prospective evaluation in randomized clinical trials, two of which are ongoing–The Appropriateness of Coronary investigation in myocardial injury and Type 2 myocardial infarction (ACT-2; ACTRN12618000378224) [26] trial

and the DEtermining the Mechanism of myocardial injury AND role of coronary disease in type 2 Myocardial Infarction (DEMAND MI; NCT03338504) trials.

Several limitations of this study should be considered. First, misclassification between patients with acute myocardial injury, chronic myocardial injury and acute MI is possible given the reliance of this study on the application of national coding rules, which may differ from clinical impression. However, this was mitigated with manual adjudication of over 6,000 EOCs and optimization of the classification criteria. Furthermore, consistency of diagnostic classification is known to be clinically very challenging especially in the setting of borderline hs-cTnT results and co-existent illnesses. In using standardized coding data and a trend towards under-diagnosis of CAD following adjudication, misclassification of encounters is likely non-differential, resulting in an underestimation of the effect estimates whilst preserving the direction of effect. Second, while we adjusted our models for key prognostic variables, residual unrecognized confounding and the impact of down-stream treatments may impact the magnitude of the observed excess hazard for various events. In part, exploring the association between troponin patterns and subsequent presentations with pneumonia and fractured neck of femur attempts to provide an evaluation of unmeasured confounding, and the lack of significant association with these recurrent events is reassuring. Lastly, we chose to use EOC-based rather than individual-level analysis allowing risk to vary dynamically with time, which may lead to correlated outcomes among frequent presenters. However, the impact of this bias is mitigated by the large available sample, the small proportion attributable to multiple repeated episodes within the same patient, and sensitivity analysis.

In conclusion, we have characterized the temporal pattern of excess hazard and potential mechanisms associated with the different phenotypes of myocardial necrosis using definitions guided by the Fourth Universal Definition of MI and high-sensitivity troponin. We observed distinct patterns of temporal risk in each group suggesting distinct therapies may be required to modify clinical outcomes. In particular, we observed a significant and additive risk for subsequent MI in patients with acute myocardial injury with CAD, suggesting potential benefit of early anatomical assessment and coronary-directed therapies. Further randomised and observational studies are required to further assess the potential role of coronary- and myocardial-targeted therapies in these populations.

## Supporting information

**S1 Fig.** Estimated hazard ratio of all-cause mortality, subsequent myocardial infarction, heart failure admission and ventricular arrhythmia or cardiac arrest in patients with A) acute myocardial infarction, B) acute myocardial injury with recognized coronary artery disease, C) acute myocardial injury without recognized coronary artery disease and D) chronic myocardial injury, relative to patients with no myocardial injury. Graphs were adjusted for age in years, sex, lowest in-hospital estimated glomerular filtration rate, maximal in-hospital high-sensitivity troponin-T, clinical comorbidities including diabetes mellitus, chronic obstructive pulmonary disease, dementia, peripheral artery disease, and previous stroke.
(DOCX)

**S2 Fig.** Estimated hazard ratio of pneumonia (left) and neck of femur fracture (right) in patients with acute myocardial infarction, acute myocardial injury with recognized coronary artery disease, acute myocardial injury without recognized coronary artery disease and chronic myocardial injury, relative to patients with no myocardial injury. Graphs were adjusted for all variables included in the flexible parametric model.
(DOCX)

**S1 Table. Observed incidence of pneumonia and neck of femur fracture at 1-year.**
CAD = coronary artery disease, NOF = neck of femur.
(DOCX)

**S2 Table. Diagnoses based on troponin pattern and diagnostic code versus adjudicated diagnoses.** CAD = coronary artery disease.
(DOCX)

**S3 Table. Estimated hazard ratios of 30-day and 1-year all-cause mortality, recurrent myocardial infarction, and subsequent heart failure admission in the acute myocardial infarction, acute myocardial injury with recognized coronary artery disease, acute myocardial injury without recognized coronary artery disease and chronic myocardial injury groups after excluding patients with 4 or more non-cardiovascular presentation per year.** The model adjusted for age in years, sex, lowest in-hospital estimated glomerular filtration rate, maximal in-hospital high-sensitivity troponin-T, clinical comorbidities such as diabetes mellitus, chronic obstructive pulmonary disease, dementia, peripheral artery disease, and previous stroke. CAD = coronary artery disease, HR = hazard ratio, 95%CI = 95% confidence interval, MI = myocardial infarction, HF = heart failure.
(DOCX)

**S4 Table. Adjusted hazard ratios of new/recurrent myocardial infarction and subsequent heart failure admission in patients with acute myocardial injury with recognized coronary artery disease, acute myocardial injury without recognized coronary artery disease and chronic myocardial injury based on multivariable Cox regression models.** Cox regression models adjusted for age in years, sex, lowest in-hospital estimated glomerular filtration rate, maximal in-hospital high-sensitivity troponin-T, clinical comorbidities including diabetes mellitus, chronic obstructive pulmonary disease, dementia, peripheral artery disease, and previous stroke. CAD = coronary artery disease, HR = hazard ratio. *This group includes both the acute myocardial infarction and the acute myocardial injury with recognized coronary artery disease groups.
(DOCX)

## Author Contributions

**Conceptualization:** Anthony (Ming-yu) Chuang, Ehsan Khan, Dylan Jones, Matthew Horsfall, Sam Lehman, Kristina Lambrakis, Martin Than, Julian Vaile, Ajay Sinhal, John K. French, Derek P. Chew.

**Data curation:** Anthony (Ming-yu) Chuang, Matthew Horsfall, Derek P. Chew.

**Formal analysis:** Anthony (Ming-yu) Chuang, Mau T. Nguyen, Ehsan Khan, Matthew Horsfall.

**Investigation:** Anthony (Ming-yu) Chuang, Mau T. Nguyen, Dylan Jones, Matthew Horsfall, Sam Lehman, Kristina Lambrakis, Martin Than, Julian Vaile, Ajay Sinhal, Derek P. Chew.

**Methodology:** Anthony (Ming-yu) Chuang, Mau T. Nguyen, Ehsan Khan, Dylan Jones, Matthew Horsfall, Sam Lehman, Nathaniel R. Smilowitz, Kristina Lambrakis, Martin Than, Julian Vaile, John K. French, Derek P. Chew.

**Supervision:** Julian Vaile, Ajay Sinhal, John K. French, Derek P. Chew.

**Writing – original draft:** Anthony (Ming-yu) Chuang, Mau T. Nguyen, Ehsan Khan, Dylan Jones, Matthew Horsfall, Sam Lehman, Nathaniel R. Smilowitz, Kristina Lambrakis, Martin Than, Julian Vaile, Ajay Sinhal, John K. French, Derek P. Chew.

**Writing – review & editing:** Anthony (Ming-yu) Chuang, Mau T. Nguyen, Ehsan Khan, Dylan Jones, Matthew Horsfall, Sam Lehman, Nathaniel R. Smilowitz, Kristina Lambrakis, Martin Than, Julian Vaile, Ajay Sinhal, John K. French, Derek P. Chew.

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
