## [Decision Letter · Decision Letter 0]

14 Jan 2021

PONE-D-20-35314

Troponin elevation pattern and subsequent cardiac and non-cardiac outcomes: Implementing the Fourth Universal Definition of Myocardial Infarction and high-sensitivity troponin at a population level

PLOS ONE

Dear Dr. Chuang,

Thank you for submitting your manuscript to PLOS ONE. After careful consideration, we feel that it has merit but does not fully meet PLOS ONE’s publication criteria as it currently stands. Therefore, we invite you to submit a revised version of the manuscript that addresses the points raised during the review process.

Your manuscript is reviewed by two reviewers and me, and found somewhat interesting. Please revise your manuscript and address all the reviewer's comments point-by-point that are given below. 

We look forward to receiving your revised manuscript.

Kind regards,

Johnson Rajasingh, Ph.D, HCLD

Academic Editor

PLOS ONE

Journal Requirements:

Reviewers' comments:

Reviewer's Responses to Questions

**Comments to the Author**

1. Is the manuscript technically sound, and do the data support the conclusions?

Reviewer #1: Yes

Reviewer #2: Yes

2. Has the statistical analysis been performed appropriately and rigorously? 

Reviewer #1: Yes

Reviewer #2: Yes

3. Have the authors made all data underlying the findings in their manuscript fully available?

Reviewer #1: No

Reviewer #2: Yes

4. Is the manuscript presented in an intelligible fashion and written in standard English?

Reviewer #1: Yes

Reviewer #2: Yes

5. Review Comments to the Author

Reviewer #1: The manuscript by Chuang et al included a large number of patients (n=372,310) and classified them into 5 groups based on high sensitivity troponin T (hs-cTnT) following the Fourth Universal Definition of Myocardial Infarction, which differentiates myocardial infarction (MI) from myocardial injury.

The major finding include the identification of about 3-fold increase in the hazard ratio for subsequent MI in patients with acute myocardial injury with recognized coronary artery disease (CAD). This increased risk for MI was not observed in patients with acute myocardial injury without recognized CAD or in patients with chronic myocardial injury.

Minor points:

The authors can try to give explanation for the finding that include a low risk for subsequent MI in patients with chronic myocardial injury.

Typo error in Discussion, 2nd paragraph, line 4

Reviewer #2: In this manuscript, the authors characterised cardiac and non-cardiac outcome patterns of myocardial infarction (MI), acute and chronic myocardial injuries per 4th definition of myocardial infarction. For this, the authors included all patients presenting to the public ED in South Australia and the episodes of care (EOC) were further categorized into 5 groups based on troponin elevation pattern and the presence of ischemia. Specifically, they focused on both short- and intermediate-term cardiovascular outcomes beyond mortality and described outcome patterns based on the groups. The study’s strength lies in the inclusiveness of all patients presenting to the public ED in South Australia which eliminates selection bias and allows health system-level study. Also, it is timely to investigate how outcome patterns of MI differ from myocardial injury based on the updated definition of myocardial infarction that was published in 2018. However, the reviewer has the following concerns:

1. Patient group classification:

a) I am not sure whether group 1 and 2 can be sufficiently differentiated based on the proposed algorithm. Specifically, group 1 includes patients with MI secondary to artherosclerotic plaque rupture. To capture these patients, the authors used troponin patterns as well as primary diagnosis of CAD (ICD code I20-I25). The use of ICD code I20-I25, however, may not be specific to capture the population of interest as it includes patients with chronic stable angina (I20) as well as chronic ischemic heart disease (I25). Additionally, it’d be useful to have more clinical information. What % of patients in group 1 vs 2 were treated with coronary revascularization, dual-antiplatelet therapies, and etc?

b) Group 4: a significant portion (8.7%) of patients fall to this group. While CAD was not coded as the secondary diagnosis, the patients in this group appear to be more likely to have known PAD and CVA. Is it possible that some of the group 4 patients were not appropriately categorized due to lack of coding?

c) Table 1: were there any pre-existing HF or arrhythmia diagnosis? For example, if patients in group 4 had pre-existing dilated cardiomyopathy, it’d explain chronic troponin elevation as well as rather higher rates of HF admission

d) Supplemental Table 2: Group 2 and 3 classifications appear to be poorly correlating with adjudication (~50% and 65% respectively)? How would this impact the overall data analysis and conclusion (since only a small fraction was adjudicated)?

2. Outcomes:

a) The authors recruited patients who presented to the ED between 2011-2019 yet only reported 30-day and 1 year outcomes. Were there any longer term outcomes available (in subset patients) and if so, were the patterns different?

The authors reported all-cause mortality. Is there information on cardiovascular mortality? It’d be interesting to know whether the mortality is primarily driven by cardiovascular death versus non-cardiac death

b) Are table 4 and figure 3 from the same analyses? If so, no need to include both

c) As the authors claim “Distinct patterns of risk were observed for each group,” I wonder whether there is any way the authors can present the data in a figure format rather than table formats to make these patterns more visible/easily recognizable.

Minor comments:

a) Figure 2: It is hard to compare how the outcome patterns differ among A-D groups. Perhaps the authors can add a final row that merges all four groups in one figure so that the comparison can be made more easily? Also wouldn’t it be better to make terminologies consistent by labeling A-D groups to group 1-4?

b) Figure 3: I assume acute MI/acute myocardial injury with recognized CAD is “pooled acute MI and acute myocardial injury with recognized CAD” per the main text? This is confusing so would suggest to clarify this as Group 1 & 2. Also, was this trend statistically significant? If so, would mark with */** based on p-values.

6. PLOS authors have the option to publish the peer review history of their article (what does this mean?). If published, this will include your full peer review and any attached files.

Reviewer #1: No

Reviewer #2: No

---

## [Author Response · Author response to Decision Letter 0]

27 Jan 2021

Dr. Joerg Heber 

Editor-in-chief 

PLOS One

18th January 2020

Dr. Heber,

We thank the three reviewers for their helpful and thoughtful suggestions, which we have used to improve our manuscript. The suggestions and criticisms made by the reviewers are addressed as follows:

Reviewer 1:

1. The major finding include the identification of about 3-fold increase in the hazard ratio for subsequent MI in patients with acute myocardial injury with recognized coronary artery disease (CAD). This increased risk for MI was not observed in patients with acute myocardial injury without recognized CAD or in patients with chronic myocardial injury. The authors can try to give explanation for the finding that include a low risk for subsequent MI in patients with chronic myocardial injury.

• We believe that acute myocardial injury and chronic myocardial injury does not necessarily denote an ischaemic event and as such may represent non-coronary events. As such, we postulated and subsequently observed in our study that if a patient has CAD, their risk of subsequent MI is significantly higher than patients without CAD. This observation was also reported in other studies (such as the SCOT-HEART and PROMISE studies) that reported the presence of coronary atherosclerosis to be associated with significantly greater risk of subsequent MI. 

2. Typo error in Discussion, 2nd paragraph, line 4

• We thank the reviewer for pointing this out and it has been corrected. 

Reviewer 2:

1. Patient group classification:

a) I am not sure whether group 1 and 2 can be sufficiently differentiated based on the proposed algorithm. Specifically, group 1 includes patients with MI secondary to artherosclerotic plaque rupture. To capture these patients, the authors used troponin patterns as well as primary diagnosis of CAD (ICD code I20-I25). The use of ICD code I20-I25, however, may not be specific to capture the population of interest as it includes patients with chronic stable angina (I20) as well as chronic ischemic heart disease (I25). Additionally, it’d be useful to have more clinical information. What % of patients in group 1 vs 2 were treated with coronary revascularization, dual-antiplatelet therapies, and etc?

• We thank the reviewer for pointing this out. We chose this method to differentiate group 1 (acute MI) versus group 2 (acute myocardial injury with CAD) as we believe any hospital presentation with an acute rise and/or fall in hs-cTnT with the ‘primary reason for presentation’ of I20-I25 represents an acute ischaemic event (i.e. acute MI based on 4th Universal Definition) even if they were labelled as having chronic ischaemic heart disease given the biomarker change. In contrast, if the ‘primary reason for presentation’ is not associated with a CAD code (I20-I25), but the patient had a recognized history of CAD, then we chose to deem these presentations as acute myocardial injury with previously recognized CAD. However, we do recognize that this is a limitation of using an administrative dataset and have previously acknowledged it in the Discussion section. Furthermore, we attempted to optimise the classification criteria and mitigate against misclassification bias by manually adjudicating a total 6,362 EOC; obtaining an accuracy of approximately 85%. We have further highlighted this limitation in the Discussion section. 

• Regarding additional clinical information, we have updated Table 1 to include coronary angiography use during index presentation. We do not have access to patient’s anti-platelet status and therefore cannot present this in our manuscript. We have acknowledged this as a limitation in the Discussion section (i.e. potential confounding due to down-stream treatments).

b) Group 4: a significant portion (8.7%) of patients fall to this group. While CAD was not coded as the secondary diagnosis, the patients in this group appear to be more likely to have known PAD and CVA. Is it possible that some of the group 4 patients were not appropriately categorized due to lack of coding?

• We agree that the possibility exists that there is under-diagnosis of CAD in our dataset. As such, we have acknowledged this as a limitation in our study. Specifically, we acknowledge the potential for “misclassification between patients with acute myocardial injury, chronic myocardial injury and acute MI”. However, for group 4, secondary diagnoses also included all known history of CAD that was documented from previous EOC for a given patient. Therefore, any previous recognised CAD would have been included and identified for an EOC even if CAD was no coded for during the EOC of interest. Therefore, this classification criteria is likely to reflect the information that is available to clinicians at the time of encounter and thus real-world clinical practice.

c) Table 1: were there any pre-existing HF or arrhythmia diagnosis? For example, if patients in group 4 had pre-existing dilated cardiomyopathy, it’d explain chronic troponin elevation as well as rather higher rates of HF admission

• We thank the reviewer for pointing this out and we agree that both HF and arrythmia are very common causes of chronic myocardial injury. We have updated Table 1 to include ‘prior known heart failure’ and ‘previous history of ventricular arrythmia.’. We would also like to point out that prior HF is a confounder that we have corrected for in our models, however, acute myocardial injury (e.g. acute decompensation or acute atrial arrythmia) or acute ischemia can still occur in patient with baseline chronic myocardial injury. 

d) Supplemental Table 2: Group 2 and 3 classifications appear to be poorly correlating with adjudication (~50% and 65% respectively)? How would this impact the overall data analysis and conclusion (since only a small fraction was adjudicated)?

• We thank the reviewer for this observation. The suboptimal correlation between the classification of groups 2 and 3 with the results of manual adjudication is likely a result of using hs-cTnT results and coded administrative dataset to classify clinical encounters. This has been acknowledged in our limitations section. The overall results of the adjudication process suggest a trend towards under-diagnosis of CAD. We believe this misclassification to likely be non-differential and therefore would result in an under-estimation of the effect estimate (i.e. biased towards neutral) whilst the direction of the effect is likely to be preserved given the size of our dataset. We have amended out manuscript to better describe this limitation. Furthermore, given the large number of clinical encounters in our study cohort, we believe it would not be possible to adjudicate all encounters.

2. Outcomes:

a) The authors recruited patients who presented to the ED between 2011-2019 yet only reported 30-day and 1 year outcomes. Were there any longer term outcomes available (in subset patients) and if so, were the patterns different? The authors reported all-cause mortality. Is there information on cardiovascular mortality? It’d be interesting to know whether the mortality is primarily driven by cardiovascular death versus non-cardiac death

• We thank the reviewer for the suggestion. Longer-term outcomes are available, but we chose 1-year outcome to ensure that we have follow-up data for our entire cohort up till year 2019. To address the review’s suggestion, we have now modelled temporal hazard out to 3 years and have presented the results in Table 3. 

b) Are table 4 and figure 3 from the same analyses? If so, no need to include both

• Thank you. We will move Table 4 into the Supplement section. 

c) As the authors claim “Distinct patterns of risk were observed for each group,” I wonder whether there is any way the authors can present the data in a figure format rather than table formats to make these patterns more visible/easily recognizable.

• We thank the reviewer for the suggestion. We have presented the data in a figure format (Figure 2). We note that the review has subsequent commented below regarding format of the figure, which we will address in the next point. 

Minor comments:

a) Figure 2: It is hard to compare how the outcome patterns differ among A-D groups. Perhaps the authors can add a final row that merges all four groups in one figure so that the comparison can be made more easily? Also wouldn’t it be better to make terminologies consistent by labeling A-D groups to group 1-4?

• We thank the reviewer for the suggestion. We have made the following changes to Figure 2:

• We have combined all four groups in one figure each for the four outcomes (death, MI, HF, and ventricular arrythmia) for Figure 2. We removed 95% CI to make the Figure more visually clear. 

• We have moved the original Figure 2 to the Supplementary section as Supp figure 1. 

b) Figure 3: I assume acute MI/acute myocardial injury with recognized CAD is “pooled acute MI and acute myocardial injury with recognized CAD” per the main text? This is confusing so would suggest to clarify this as Group 1 & 2. Also, was this trend statistically significant? If so, would mark with */** based on p-values.

• We thank the reviewer for the suggestion. We have updated Figure 3 with new labelling and statistical testing markers. 

Again, we appreciate the valuable reviews that have helped to improve the manuscript. Thank you and we look forward to hearing from you soon.

Kind Regards,

---

## [Decision Letter · Decision Letter 1]

24 Feb 2021

Troponin elevation pattern and subsequent cardiac and non-cardiac outcomes: Implementing the Fourth Universal Definition of Myocardial Infarction and high-sensitivity troponin at a population level

PONE-D-20-35314R1

Dear Dr. Chuang,

We’re pleased to inform you that your manuscript has been judged scientifically suitable for publication and will be formally accepted for publication once it meets all outstanding technical requirements.

Kind regards,

Johnson Rajasingh, Ph.D, HCLD

Academic Editor

PLOS ONE

Additional Editor Comments (optional):

Reviewers' comments:

Reviewer's Responses to Questions

**Comments to the Author**

1. If the authors have adequately addressed your comments raised in a previous round of review and you feel that this manuscript is now acceptable for publication, you may indicate that here to bypass the “Comments to the Author” section, enter your conflict of interest statement in the “Confidential to Editor” section, and submit your "Accept" recommendation.

Reviewer #1: All comments have been addressed

Reviewer #2: All comments have been addressed

2. Is the manuscript technically sound, and do the data support the conclusions?

Reviewer #1: Yes

Reviewer #2: Yes

3. Has the statistical analysis been performed appropriately and rigorously? 

Reviewer #1: Yes

Reviewer #2: Yes

4. Have the authors made all data underlying the findings in their manuscript fully available?

Reviewer #1: Yes

Reviewer #2: No

5. Is the manuscript presented in an intelligible fashion and written in standard English?

Reviewer #1: Yes

Reviewer #2: Yes

6. Review Comments to the Author

Reviewer #1: The authors have addressed all the issues and revised the manuscript based on the criticisms from the reviewers.

Reviewer #2: The authors have sufficiently addressed my comments. Please see below for minor edits/errors:

1) Table 3: Recurrent MI 3 year HR should be 3.43-5.42 (not 34.3)

2) For figures: please add x and y-axis (assume x is HR and y is days of outcome but this information is missing in many of the figures)

3) Figure 3: please harmonize figure/font size between figure 3(a) and (b)

7. PLOS authors have the option to publish the peer review history of their article (what does this mean?). If published, this will include your full peer review and any attached files.

Reviewer #1: No

Reviewer #2: No

---

## [Editor Report · Acceptance letter]

26 Feb 2021

PONE-D-20-35314R1 

Troponin elevation pattern and subsequent cardiac and non-cardiac outcomes: Implementing the Fourth Universal Definition of Myocardial Infarction and high-sensitivity troponin at a population level 

Dear Dr. chuang:

I'm pleased to inform you that your manuscript has been deemed suitable for publication in PLOS ONE. Congratulations! Your manuscript is now with our production department. 

Kind regards, 

on behalf of

Dr. Johnson Rajasingh 

Academic Editor

PLOS ONE